# Minimal impact of model biases on northern hemisphere ENSO teleconnections.

Nicholas L. Tyrrell[1], Alexey Yu. Karpechko[1]

[1]Finnish Meteorological Institute, Helsinki, 00500, Finland

*Correspondence to*: Nicholas L. Tyrrell (nicholas.tyrrell@fmi.fi)

**Abstract.** Correctly capturing the teleconnection between the El Niño–Southern Oscillation (ENSO) and Europe is of importance for seasonal prediction. Here we investigate how systematic model biases may affect this teleconnection. A two–step bias–correction process is applied to an atmospheric general circulation model to reduce errors in the climatology. The bias–corrections are applied to the troposphere and stratosphere independently and jointly to produce a range of climates.

ENSO type sensitivity experiments are then performed to reveal the impact of differing climatologies on the ENSO–Europe teleconnections.

The bias–corrections do not affect the response of the tropical atmosphere nor the Aleutian Low to the strong ENSO anomalies imposed in our experiments. However, in El Niño experiments the anomalous upward wave flux and the response of the northern hemisphere polar vortex differ between the climatologies. We attribute this to a reduced sensitivity of the

upward wave fluxes to the Aleutian Low response in the bias-correction experiments, where the reduced biases result in a deepened Aleutian Low in the base state. Despite the differing responses of the polar vortex, the NAO response is similar between the climatologies, implying that for strong ENSO events the stratospheric pathway may not be the dominant pathway for the ENSO–North Atlantic teleconnection.

## 1 Introduction

The El Niño–Southern Oscillation (ENSO) has been shown to influence European climate via tropospheric and stratospheric teleconnections. Although ENSO is a key driver of global variability on seasonal to annual timescales, it's effect on Europe is less robust (Brönnimann 2007), and exhibits decadal variability (Rodríguez–Fonseca et al., 2016). The large seasonal variability in the mid–latitude Northern hemisphere, and relatively low number of observed ENSO events create some difficulty in measuring the effect in observational data. The ENSO–Europe teleconnection begins with anomalous

convection in the tropical Pacific, and during El Niño events this leads to increased divergence in the upper troposphere, creating a Rossby wave source (Hoskins and Karoly, 1981). The anomalous Rossby waves propagate to the Northern Pacific where they strengthen the wintertime Aleutian Low. There are multiple possible connections between the North Pacific anomalies and the North Atlantic (Jiménez–Esteve and Domeisen, 2018), with a tendency for a negative North Atlantic Oscillation (NAO) during El Niño events. For the stratospheric connection, as reviewed by Domeisen et al., (2019), the

deepened Aleutian low can lead to upward propagating waves, particularly of wavenumber 1, which travel into the stratosphere and weaken the wintertime stratospheric polar vortex. For strong vortex weakening events, such as sudden stratospheric warmings, anomalies can propagate down to the troposphere and project onto the Northern Annular Mode, and the NAO (Butler et al., 2014). Mezzina et al. (2020) suggest the NAO-like patterns that result from ENSO variability are distinct from the NAO, and results from a Rossby wave train from the tropics to the North Atlantic which does not affect

NAO variability. Some aspects of the teleconnection are approximately opposite for La Niña events; there is reduced upper tropospheric divergence and resultant Rossby wave, and a shallower Aleutian Low, but the anomalous response is weaker with a less consistent extratropical response (e.g. Jiménez–Esteve and Domeisen, 2019).

Both the tropospheric and stratospheric teleconnection pathway can be simulated with climate models of sufficient resolution (Cagnazzo and Manzini, 2009, Bell et al. 2009). Models also allow for large numbers of ENSO events to be simulated,

which has revealed non–linearities in teleconnections (Frauen et al., 2014, Jiménez–Esteve and Domeisen, 2019, Garfinkel et al., 2019). However, for confidence in modelling results, we need an understanding of the deficiencies of models. A fully coupled model needs to correctly represent both the complex dynamics of the ENSO ocean–atmosphere interactions to generate the convective anomalies that drive the teleconnections, and the mean climatology so the anomalies interact with the base state correctly. For example, the convective response of tropical Pacific is dependent on the mean state of the

Walker circulation (Bayr et al. 2018). The location and strength of the convective response is then important in controlling the location of the extratropical pressure response (Bayr et al., 2019), and can lead to non–linearities. In addition to the patterns of climatological SSTs, the state of the tropical (e.g. Quasi-Biennial Oscillation (QBO) phase) and extratropical atmosphere can influence the response of the NAO and polar vortex (Garfinkel et al., 2007), and biases in the subtropical jet can affect the propagation of Rossby waves from the tropics to the extratropics (Li et al., 2020)

The technique of flux correcting SSTs has been used to study the effect of model biases on ENSO dynamics (Spencer et al., 2007, Manganello and Huang, 2009, Dommenget et al., 2014) and seasonal forecasting (Magnusson et al., 2013a, Magnusson et al., 2013b). Empirical corrections are added to the coupling between the ocean and atmosphere to push the model towards the observed climatology. It is possible to use a similar technique on the prognostic atmospheric variables of a model. This bias–correction technique was used by Kharin and Scinocca (2012), and artificially decreased biases were

associated with an increase in predictive skill on seasonal timescales. Simpson et al., (2013a, 2013b) used the technique to study the impact of jet latitude bias on the Southern Annular Mode (SAM), although they did not see improvements in the persistence of the SAM when they reduced biases in the jet. When Chang et al. (2019) used a similar bias–correction technique they found an improvement in the North Pacific jet and North American rainfall climatology, and a modest improvement in seasonal forecast skill. Tyrrell et al., (2020) investigated how climatological biases affect the relationship

between the Eurasian snow extent and the wintertime polar vortex, and found that the strength of the vortex had only a small effect on its response to a tropospheric forcing, however, the downward propagation of stratospheric anomalies was sensitive to the tropospheric circulation.

     In this study we have used a similar bias correction technique to probe the impact of climatological biases on the communication of ENSO anomalies from the tropical Pacific to the North Atlantic and European sector.  The technical

details of the model and experiments are described in section 2, the results of the bias corrections and ENSO experiments are described in section 3, and a discussion and conclusions are presented in section 4.

## 2 Data and Methods

     We use the ECHAM6 spectral atmospheric model (Stevens et al., 2013), run with a horizontal truncation of T63 and 95 vertical levels with a model top at 0.02 hPa. The bias correction technique follows Kharin and Scinocca, 2012, and is similar

to that described in Tyrrell et al., 2020 (T20). It is a two–step process; first, the bias correction terms are calculated in a nudged training stage. The model's prognostic variables – divergence, vorticity, temperature, and log of surface pressure – are all nudged towards ERA–Interim for 30 years, and the nudging tendencies are recorded every 6 hours. Then the nudging tendencies for the divergence and temperature are used to create a year-long climatology of correction terms. This climatology is then smoothed in time with a Gaussian filter with a 25 day window, and it represents the inherent bias in the

model's prognostic variables. Secondly, the divergence and temperature correction terms are added to the free running model as an additional tendency term at each time step. An important difference between the nudged and bias corrected runs is that the bias correction terms are independent of the current model state, so the model can respond to perturbations, whereas during the nudged run the model is tightly constrained to the reanalysis. The technical details of the bias correction are outlined in T20, with two differences for the current experiments. For the training step the model was nudged to ERA–

Interim data from 1979–2009, whereas in T20 only the years 1979–1989 were used. The resulting bias correction terms were very similar and this did not impact the results. The second difference to T20 was that the only bias correction terms used for

this study were the divergence and temperature, rather than the divergence, vorticity, temperature and log of surface pressure. During the training stage all the model's prognostic variables were nudged towards ERA Interim, and it was found that using only two of the temperature, divergence and vorticity of the bias correction terms gave the best results for reducing the biases in the winds and temperature. Through testing different combinations we found that bias correcting only the divergence and temperature lead to the biggest decreases in the climatological biases of the control run.

The bias corrections were applied on model levels between approximately 850 hPa and 2.6 hPa, and three types of bias correction runs were performed; the troposphere and stratosphere were corrected in FullBC, the stratosphere only in StratBC, and the troposphere only in TropBC (see Table 1 for details). Then ENSO SST forcing experiments were conducted with each of these bias corrected climatologies. To generate the SST pattern we used a regression of the Niño3.4 timeseries and HadISST SSTs from 1979–2009. Only positive regression values between 30°S and 30°N and east of 150°E in the Pacific Ocean were used for the pattern, and the regression values were multiplied by 1.5 to strengthen the response, which corresponds approximately to an El Niño or La Niña forcing magnitude of 1.5K. Climatological SSTs, using HadISST data from 1979-2009, were used for the control (CTRL) run. The ENSO pattern was added to (El Niño), or subtracted from (La Niña) the SST climatology in the tropical Pacific, with climatological SSTs used everywhere else. The ENSO anomaly pattern was kept constant in time, i.e., the anomaly did not vary seasonally relative to the climatological SSTs. Each experiment was run for 100 years. It should be noted that using a regression to generate ENSO patterns results in symmetric El Niño/La Niña magnitudes, whereas from observations El Niño anomalies tend to be larger than La Niña and have a slightly different structure. This simplification, along with a constant ENSO forcing and using climatological SSTs outside the Pacific Ocean basin, has the advantage of reducing the number of controlling parameters when analysing the results of the bias-corrections, which was the main aim of the research. However, the simplifications should be considered when comparing the results with observations, particularly in relation to the intra-seasonal and early winter ENSO-Atlantic connection (e.g. King et al., 2018) that may be driven by SSTs and rainfall away from the Pacific (Ayarzagüena et al., 2018, Abid et al., 2021).

To calculate the biases between the model and reanalysis (Figure 1) we use ERA Interim 1979–2009, since that data was used to train our bias correction scheme. However, when analysing the response to El Niño and La Niña runs we use the newer ERA5 data (Hersbach et al., 2020), including the ERA5 Back Extension (Bell et al., 2020), from 1950 to 2021. To composite the data the NINO3.4 index was used for DJF (https://origin.cpc.ncep.noaa.gov/products/analysis_monitoring/ensostuff/ONI_v5.php). El Niño years defined as NINO3.4 above 0.9 K (13 years), La Niña years NINO3.4 below -0.9 K (16 years) and the years between -0.5 K and 0.5 K were defined as neutral years (19 years). The slightly stricter threshold of +/- 0.9 K was used to define the El Niño/La Niña years to include only stronger events. The years included in the ERA5 composite are listed in Supplementary Table 1.

## 3 Results

### 3.1 Reduced model biases

The climatological biases of the model's wind and temperature vary with latitude, height and season. Here we focus on the extended winter season (November-March, NDJFM) because this is the season when the ENSO teleconnection to the Northern Hemisphere is the strongest. In Figure 1 we show the mean NDJFM biases in the zonal mean zonal wind, zonal temperature and mean sea level pressure. The biases are calculated as 100 years of the model climatology minus the 1980–2009 ERA Interim climatology. The largest tropospheric biases in UZ in the CTRL (Figure 1 a) are associated with the

subtropical jet, which is too poleward, but in general the UZ and temperature biases are small in the troposphere, in particular in the tropical troposphere. Figure 1 a and e show that the stratospheric vortex is too weak and warm in the CTRL, and the cold bias at 200hPa (Figure 1 e) in the high latitudes in indicative of a too high extratropical tropopause. The sea level pressure (SLP) biases in the control run (Figure 1 i) show an annular pattern of low pressure around 60° to 80°N, and high pressure to the south of that over Africa and Asia and the north Pacific, which results in a weak Aleutian Low.

Supplementary Figure 1 shows that the spatial pattern of biases in the geopotential height at 300hPa is very similar to that in SLP suggesting that the biases are nearly barotropic.

The bias corrections are applied at different model levels, hence, the biases in control run are reduced to different extents in the three bias corrected runs. The bias in the subtropical jet is reduced in the FullBC (Figure 1 b) and TropBC (Figure 1 d) runs but not in StratBC (Figure 1 c). In the stratosphere the bias towards a too weak and too warm stratospheric polar vortex

is almost completely removed in FullBC, and reduced in StratBC. In TropBC the bias in stratospheric UZ is worsened slightly (Figure 1 d), however, the stratospheric temperature bias is slightly improved compared to CTRL (compare Figure 1 e and h). The tropopause temperature bias in the high latitudes is reduced in FullBC and TropBC, at the expense of introducing cold anomalies in the tropical upper troposphere (Figure 1 f, h). The Aleutian Low anomaly is reduced in FullBC (Figure 1 j) and TropBC (Figure 1 l), but not in StratBC (Figure 1 k), and the reduction in bias, like the bias itself, is

barotropic as shown in Supplementary Figure 1. Note that the strong westerly bias in the equatorial stratosphere is not corrected. This bias is associated with the Quasi-Biennial Oscillation (QBO), internally generated in our model, and the lack of bias correction is partly due to the fact that our approach with annually varying bias-correcting tendencies is not optimized for the QBO, as discussed in Karpechko et al. (2021).

Overall, the bias correction technique is effective at reducing biases throughout the atmosphere, and the different bias

correction experiments allow us to isolate biases in various atmospheric features such as the polar vortex, subtropical jet and the Aleutian Low, which are relevant for ENSO teleconnections to the high latitudes.

### 3.2 Teleconnection response to ENSO

We trace the path of ENSO anomalies from the tropical Pacific to the northern hemisphere polar regions and the North Atlantic. Following from Fig. 3 in Jiménez–Esteve and Domeisen, 2019, our Figure 2 shows the anomalous response of

indices chosen to highlight the ENSO teleconnection to the North Atlantic. The El Niño and La Niña forcing does not vary seasonally in our experiments, thus is not shown. ERA5 values are included for reference (see Supplementary Table 1 for the years included in the ERA5 composites), although direct comparison with the model runs is difficult due to the idealized experimental setup. The convective response of the tropical atmosphere to SST anomalies is represented by the meridional divergent wind at 100 hPa defined in the region 0°–20°N, 160°–220°E (Figure 2 a). As expected, the positive anomalies for El Niño are greater than the negative La Niña anomalies, however, the ERA5 anomalies are more symmetric for La Niña. We also see there is no significant difference between the bias-corrected experiments. This is not surprising given the small biases in the tropical troposphere, and reasonably small improvements in the tropics between the control and bias corrected experiments (Figure 1). The anomalous divergence creates a Rossby wave that leads to a deepening (El Niño) or weakening (La Niña) of the Aleutian low. We measure this using an Aleutian Low Index (ALI), defined as the SLP between 35°–60°N, 180°–240°E, indicated by the green box in Figure 1 i. The response of the ALI is proportional to the tropical divergence, with the anomalous negative El Niño response being greater than the positive La Niña response. Again, there are no clear differences in the anomalous response between the different climatologies. However, in contrast to the tropical regions, the FullBC and TropBC runs have reduced Aleutian Low biases compared to the CTRL and StratBC runs (Figure 1 i–l), implying that the response of the Aleutian Low to an ENSO signal is not dependant on model biases. The modelled ALI anomalies have a greater magnitude than ERA5.

The next step in the teleconnection is the response of heat flux at 100 hPa, 45°–75°N (HF). The anomalous HF for both El Niño and La Niña shows differences between the bias corrected runs. For an El Niño forcing, the CTRL and StratBC runs show an increase in HF with significant values (indicated by black crosses) for the DJF and JFM three–month means, whereas the FullBC and TropBC anomalies are about half as strong and have no significant values. For the La Niña forcing all the models show a negative anomalous HF, with the absolute value of the anomaly being weaker than the El Niño response. The positive HF response to El Niño in the FullBC and TropBC was about half as strong as the CTRL and StratBC response, whereas for the La Niña response the CTRL and TropBC anomalous negative HF was around half the value of the StratBC and FullBC runs. There were no significant HF values at the 5% level for any of the La Niña experiments. The lack of significance could be partly due to the weaker Rossby wave source associated with a La Niña, and the high variability of the HF. The response of HF in ERA5 exhibits more variability than that in the model. The ERA5 HF El Niño anomalies peak in JFM, where they are larger than in the all model runs. La Niña HF anomalies in ERA5 are negative in early winter and change to positive in late winter, with the late winter positive response being of opposite sign to what is seen in our model. To test if this was due to sampling uncertainty in observations we subsampled our model for periods of 16 years, to match La Nina events in ERA5, and the subsampling was performed 500 times, as shown in Supplementary Figure 2. The HF was used since this was a controlling factor in the stratospheric response, and the HF value for Jan-Feb-Mar is shown, since this differed the most between ERA5 and the model. It was found that the observed values of the La Niña HF are within the sampling uncertainty of all model runs, indicating that there is no evidence that the models do not capture the observed stratospheric pathway of ENSO teleconnections.

We measure the response of the stratospheric polar vortex with the zonal mean zonal wind at 60N and 10 hPa (UZ60). UZ60
is well predicted by the HF values. Namely, for an El Niño forcing the CTRL and StratBC experiments show a significant
weakening, whereas the response of FullBC and TropBC only show a slight weakening of the vortex (Figure 2 d). Likewise
with the La Niña forcing the FullBC and StratBC show a greater strengthening than the CTRL and TropBC. These results
are also true for the lower stratosphere as measured with the zonal mean zonal wind at 60N and 100hPa (Figure 2 e). The
seasonal mean of the full zonal mean wind response is shown in Supplementary figure 3. As with the ALI response, the
response of the polar vortex does not appear to be affected by biases in the strength of the vortex, and is instead fully
explained by the heat flux response, which is discussed further in the next section. Both the El Niño and La Niña response in
ERA5 changes sign throughout the extended winter season at both 10hPa and 100hPa, which is not seen in the models.
However, the observed values are not inconsistent with zero in any season, suggesting that the observational records may be
too short to constraint the sign of the stratospheric pathway of the ENSO teleconnection.

The differences in the magnitude of the stratospheric responses are not mirrored in the response of the NAO, despite the
well-known connection between the vortex and the NAO and the importance of a realistic stratosphere for the ENSO–North
Atlantic teleconnection (for example, as shown by Cagnazzo and Manzini, 2009, with an older version of the model used in
our study). There is a weaker FullBC response in early winter UZ60 to La Nina, and correspondingly weaker NAO response
in early winter. However, for the other runs the strength of the polar vortex anomaly has no clear connection with the
response of the NAO. This is evident when comparing the El Niño response of the CTRL and StratBC, to FullBC and
TropBC, where the latter two have a weak UZ60 response but a strong NAO response. The NAO response to El Niño in
ERA5 is also negative but weaker than the models, and the ERA5 NAO response shows statistical significance which is not
seen in the stratospheric indices for ERA5. ERA5 also shows a weak insignificant positive NAO during mid and late winter.

To determine the reason for the weak connection between the stratospheric anomalies and the NAO in the models, we
investigate scatter plots of uz 60N 10hPa and the NAO index, as shown in Figure 3. For this figure we chose to show the
variability within each experiment (i.e. each of 100 years of DJF means for El Niño, neutral and La Niña, and for ERA5, 13
El Niño years, 16 La Niña years and 19 neutral years) to better understand the time-mean sensitivity of the ENSO
teleconnection, as well as the sensitivity of the stratosphere-troposphere couplings within different bias corrected runs. The
positive regression slopes in Figure 3 show that there is the expected relationship between UZ60 and the NAO for each year
within each experiment – that is, a weaker vortex (lower values of UZ60) is associated with a more negative NAO. For each
model the large crosses in Figure 3 show the mean value of El Niño years (red crosses), La Niña years (blue crosses) and
neutral years (green crosses). There is an indication that the positive correlation between vortex strength and NAO is also
apparent between the mean values of the El Niño, and neutral/La Niña years, with a much smaller signal between the neutral
and La Niña experiments. However, the large variability within each experiment means that the ENSO-forced difference is
relatively small. Although causality is not explained, the figure demonstrates that the stratosphere-troposphere coupling does
not play a dominant role in the ENSO-EU teleconnection in our experiments. The effect of a weaker or stronger vortex on
the NAO is relatively small compared to variability, and hence, the different polar vortex responses between the bias

correction experiments do not translate neatly into different magnitudes of the NAO response. One could also hypothesize that a weaker polar vortex response to El Niño in FullBC and TropBC may cause an NAO response of similar magnitude to that in CTRL and StratBC (Figure 2 f) because of an increased sensitivity of NAO to the stratospheric variability. However, the similarity of the UZ60-NAO correlation coefficients between the experiments does not support this hypothesis. Instead, it reveals that the sensitivity of the NAO to the stratospheric variability remains unchanged by the bias corrections. Figure 3 a shows the NAO and UZ60 for ERA5, divided into ENSO phases. The behaviour is broadly similar to the model, although the regressions for both El Niño and La Niña are not statistically significant. Overall, for our experiments with a strong ENSO forcing, the stratosphere plays only a minor role in the NAO response.

### 3.3 A mechanism for differences in the simulated polar vortex El Niño response

To investigate the cause of the different HF response between the experiments it is necessary to consider the effect of the bias corrections. To do that we must consider the absolute values and anomalous response together. In Figure 4 we show the DJF ALI, HF and UZ60, for La Niña, neutral and El Niño conditions. ERA5 is included to demonstrate the biases. Figure 4a shows the reduced bias in the ALI in the FullBC and TropBC experiments compared to the CTRL and StratBC experiments. The figure also shows the deepening and weakening of the ALI for the ENSO forcing is fairly constant between the bias correction runs, hence, the larger biases in the CTRL and StratBC experiments do not impact the response of the Aleutian low. In Figure 4b we again see the reduced bias in the FullBC and TropBC experiments for the HF. However, there is a smaller change between neutral and El Niño conditions for the FullBC and TropBC compared to the CTRL and StratBC. In other words, in the model the HF is less sensitive to the deepening of the Aleutian low in the bias corrected runs where the Aleutian low is already deeper due to the reduced bias. In ERA5 the Aleutian low is deeper, yet responds more like the biased runs, although we caution against using this to dismiss the model results, due to the possibility of sampling error (e.g. Supplementary figure 2).

As shown in Figure 2d, the response of the polar vortex is controlled by the HF response in the different climatologies. Figure 4c reiterates this, and also demonstrates that the biases in the polar vortex do not impact its response to anomalous wave forcing. In neutral conditions the UZ60 biases in the StratBC and FullBC are around 5 m s$^{-1}$ less than the CTRL and TropBC biases. However, for El Niño conditions the FullBC (less bias, stronger vortex) and TropBC (more biased, weaker vortex) have a weak response, and the StratBC (less bias, stronger vortex) and CTRL (more biased, weaker vortex) have a strong response.

We have shown that the stratospheric response to an El Niño forcing is partially dependant on model biases, and seems to be related to the sensitivity of the HF to the Aleutian low. In Figure 5 we use regressions of HF onto SLP to show how the effectiveness of wave forcing by the Aleutian low changes between neutral and El Niño conditions. Similar to Figure 3, we again consider the variability within each bias-correction and ENSO experiment, to understand the time-mean response. Figure 5 a–d shows the regression of monthly HF onto monthly SLP for the extended winter months (November–March) in neutral ENSO conditions. The areas of SLP that are most strongly associated with HF are the Aleutian low region and

Siberia, with weaker connections over Greenland and North America. In neutral years these features are very similar between the control and the bias corrected runs, which means the deeper Aleutian low has not affected its association with wave driving. Figure 5 e–h is the same regression in years with an El Niño forcing. Rather than testing the difference between neutral and El Niño years, this is now a measure of variability during El Niño years. The connection between SLP in the Aleutian low and HF is lower in the CTRL and StratBC El Niño runs, but is now absent in the FullBC run and very weak in the TropBC runs. Therefore, for an equally large Aleutian low anomaly, there would result in less wave forcing in the FullBC and TropBC. There appears to be a threshold for the depth of the Aleutian Low, below which any additional anomalies do not result in additional wave forcing. During La Niña years the regression values over the Aleutian low region are slightly stronger in the FullBC and TropBC, but this did not lead to differences in the response of the HF (i.e. Figure 4b, La Niña to neutral changes).

In Figure 6 the HF was plotted against the Aleutian low SLP anomalies to look for a non-linear saturation in the wave forcing by the Aleutian Low. For variability within the experiments there is an indication that the relationship between HF and Aleutian low breaks down with low Aleutian low values, as indicated by the steeper regression line for La Nina years (blue) and flatter line for El Niño years (red). For the FullBC and TropBC El Niño experiments the relationship actually reverses slightly, shown by the positive regression in Figure 6 c and e (red line). ERA5 does not show changes to the HF-Aleutian low between the ENSO composites, however, the correlations are weak so it is difficult to make conclusions based on that data. Although Figure 6 does not conclusively explain the differences between the bias correction experiments, it shows for our model that the HF-Aleutian low relationship does change in tandem with the absolute value of the Aleutian low, and the behaviour is fairly consistent within each set of bias-correction experiments for different ENSO forcings.

## 4 Discussion and Conclusions

By applying bias correction terms to the divergence and temperature tendencies of an atmospheric model we have reduced biases in the tropospheric and stratospheric mean states to create various climatologies. Within the different climatologies we have performed idealized ENSO forcing experiments to test the role of biases in ENSO teleconnections. There were only small reductions in the bias in the tropics, and there was no difference in the convective response to ENSO between the bias correction experiments. Likewise, the anomalous response of the Aleutian low to El Niño and La Niña forcing was similar between the experiments, despite reduced biases in the Northern hemisphere extratropical sea level pressure. Li et al (2020) showed an equatorward jet bias can dampen the response of the Aleutian Low to a tropical Rossby wave source. The jet in the CTRL run has a slight poleward bias (Figure 2 a), which is improved in the FullBC and TropBC runs, but this did not affect the response of the Aleutian low. Hence, reductions in the subtropical jet biases for our model is likely not important for the ENSO teleconnection. We find that reducing certain climatological biases in the surface pressure and wind speed does not significantly affect the response of the Aleutian low to Rossby wave forcing. Reduced biases in the Aleutian low SLP did, however, lead to differences in the anomalous upward wave flux associated with a deepened low due to an El Niño

forcing, so the model's ability to generate a planetary wave flux may be dependent on biases in surface pressure. The response of the polar vortex was shown to be dependent on the upward planetary wave forcing, and not affected by local biases in the strength of the vortex. A stronger polar vortex in the experiments with stratospheric bias corrections did not affect the anomalous response of the vortex to wave forcing. The NAO response was shown to not be sensitive to the stratospheric representation nor the stratospheric response, we conclude that in our experiments - with an SST forcing that corresponding to large ENSO events - it is dominated by tropospheric teleconnections. This result appears consistent with Bell et al. (2009) who also found that for strong ENSO events the tropospheric teleconnections dominate.

To validate our model results we compare them with ERA5 reanalysis from 1950 to 2021. A threshold of 0.9K in the NINO3.4 region was used to composite large ENSO events in the reanalysis, resulting in 13 El Niño and 16 La Niña events (all years listed in Supplementary Table 1). When comparing the model results to ERA5 the main differences occurred for the HF and stratospheric ENSO response. ERA5 exhibits more variability throughout winter which is not present in the model. However, our tests suggested that sampling errors in the reanalysis (see Supplementary figure 2) may be the reason for the difference. Note that in order to simplify comparison between model experiments we applied a simple constant ENSO SST forcing in all model runs, which is not an optimal approach for comparison with observations. A more realistic experimental setup would be better suited to examine the differences with observed teleconnections. Also, we note that despite the variability of the stratospheric response in ERA5, the observed NAO response was more similar to the model, which again points to dominant role of tropospheric teleconnections for strong El Niño events.

Although one motivation behind artificially bias correcting the model was to investigate how the response to various forcings might improve if the biases were reduced, it should be noted that the ECHAM atmospheric model has already been shown to have a realistic response to an ENSO forcing (Manzini et al. 2006, Cagnazzo and Manzini, 2009). The ENSO teleconnection, the Siberian snow–polar vortex connection investigated in Tyrrell et al. (2020), and the QBO teleconnection investigated in Karpechko et al. (2021) are all relevant to seasonal forecasting, but the bias correction technique is unlikely to be used for operational forecasting. Hence, for these experiments the bias corrections are a tool that is used not necessarily to improve the response relative to observations, but rather to explore the sensitivity of the response to climatological biases.

It was interesting to find that response of the Aleutian low and the stratospheric polar vortex was not affected by the climatological biases that we reduced. These two features are important in the ENSO–to–Europe teleconnection and had large reductions in biases due to the corrections. These features are also both forced by planetary waves; horizontally propagating waves from anomalous convection in the tropical Pacific, or vertically propagating waves from the northern hemisphere troposphere to the stratosphere. Hence, model biases in the depth of the Aleutian low, or the magnitude of the polar vortex winds, do not appear to strongly affect their response to wave forcing.

The control and bias corrected runs differed in the magnitude of wave forcing caused by the deepening Aleutian low due to the El Niño forcing. We theorize that this was due to the relationship between the depth of the Aleutian low and its effectiveness at wave forcing. The two experiments with bias corrections in the troposphere both had a deeper Aleutian low, which was closer to observations. Although the magnitude of ALI anomaly was the same, the runs with a deeper Aleutian

low had reduced wave forcing for El Niño conditions. Regressions between SLP and HF showed that lower Aleutian low SLP was associated with a decreasing correlation between Aleutian low SLP and HF. Therefore, we speculate that the reduced wave forcing when the troposphere was bias corrected in the FullBC and TropBC, was due to the lower climatological SLP values in the Aleutian low area. It appears that at some threshold of low values of SLP, further anomalies in the Aleutian low do not result in anomalous upward waves. The opposite was not true for the La Niña conditions, since there appears to be no maximum values where the relationship between HF and Aleutian low SLP changes. Additionally, the non–linearity in the El Niño/La Niña atmospheric response (e.g. Frauen et al., 2014) means that the La Niña response is smaller, making it more difficult to distinguish robust differences between the climatologies. By artificially bias correcting an atmospheric model, we have shown that some aspects of ENSO teleconnections are very robust to the specific model biases we corrected, while more subtle interactions of anomalies with the basic state can impact the overall response. A deeper understanding of the influence of inherent model biases on teleconnections can guide future model development, and also aid in the physical understanding of these important teleconnections.

**Data availability**

The climatological means of all model experiments, for the variables used in this paper are available at: https://figshare.com/articles/dataset/ECHAM6_Bias_Correction_ENSO/13311623. The full timeseries is available upon request to Nicholas Tyrrell. ERA-Interim and ERA5 data Copernicus Climate Change Service Climate Data Store (CDS, https://climate.copernicus.eu/climate-reanalysis). The ECHAM6 model is available to the scientific community under a version of the MPI-M license http://www.mpimet.mpg.de/en/science/models/license/. The HadISST SST and sea ice data are available from the U.K. Met Office https://www.metoffice.gov.uk/hadobs/hadisst/.

**Author contributions**

NLT conducted the model runs, analysis, and wrote the first draft. AYK contributed to the interpretation of the results, and improving the final manuscript.

**Competing interests**

The authors declare that they have no conflict of interest.

**Acknowledgements**

The authors would like to acknowledge Sebastian Rast, John Scinocca, Slava Kharin, and Michael Sigmond for invaluable technical and scientific help. N. L. T. and A. Y. K. are funded by the Academy of Finland (Grants 333255, 286298, and 294120). Two anonymous reviewers greatly helped to improve the manuscript.

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

**Table 1. Experiment details and run names.**

| Bias corrections | ENSO neutral | El Niño | La Niña |
|---|---|---|---|
| None | CTRL | CTRL_EN | CTRL_LN |
| 850 hPa to 2.6 hPa | FullBC | FullBC_EN | FullBC_LN |
| 100 hPa to 2.6 hPa | StratBC | StratBC_EN | StratBC_LN |
| 850 hPa to 100 hPa | TropBC | TropBC_EN | TropBC_LN |



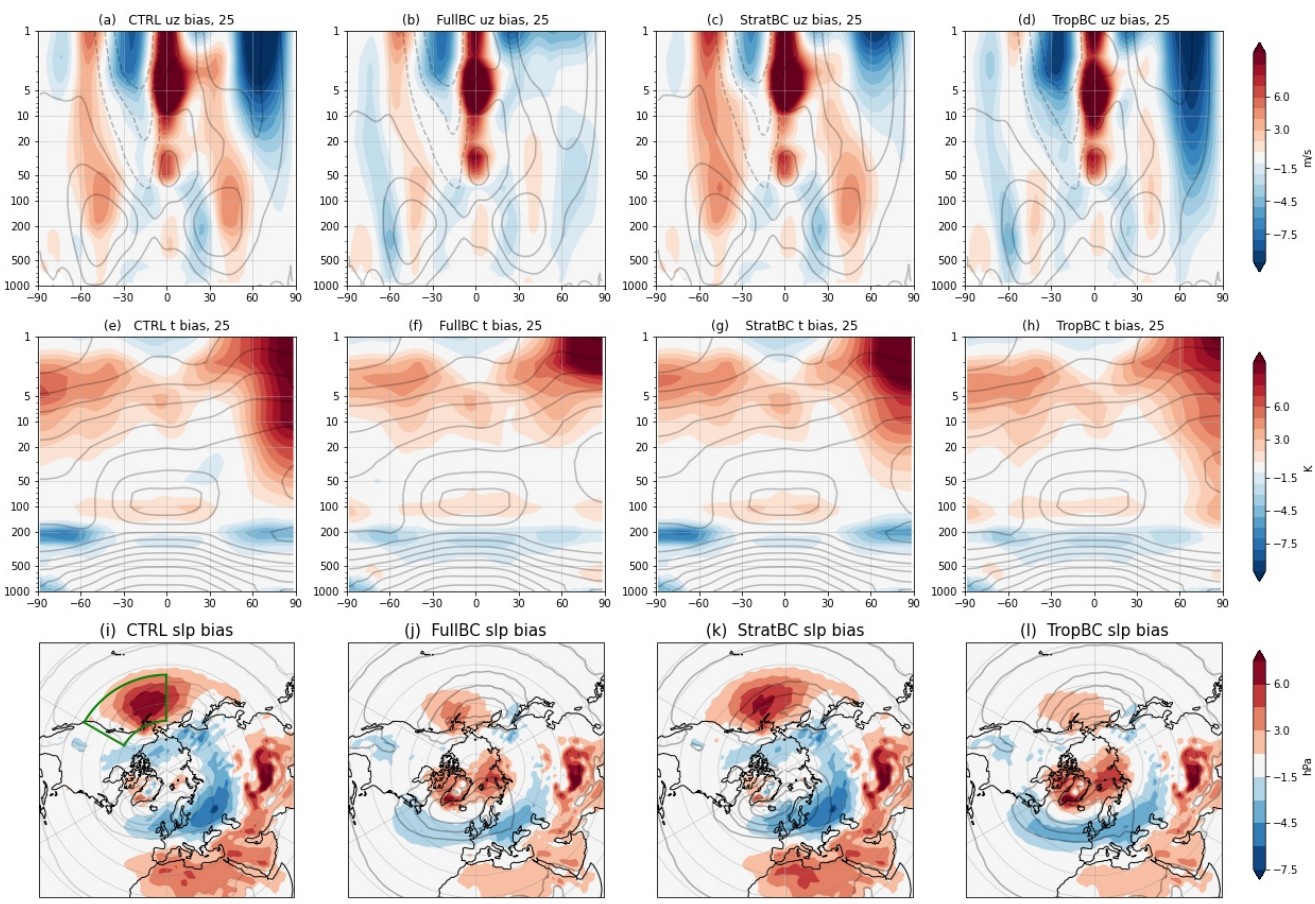

**Figure 1: Nov-Mar bias of zonal mean zonal wind (top row), zonal mean temperature (middle row) and MSLP (bottom row) in the four experiments; CTRL (a, e, i), FullBC (b, f, j), StratBC (c, g, k), and TropBC (d, h, l). The bias is calculated as the difference between model and ERA Interim climatology (1979-2009). Grey contours show model climatology. Negative winds in the top rows are marked with dashed lines. Contours are drawn at intervals of 10 m/s for zonal winds (-30, -20, ..., 30 m/s), at 10K for temperatures (200, 210, ..., 290K), and at 5 hPa for SLP (990, 995, ..., 1030hPa). The green box in (i) shows the area of the Aleutian Low Index used in Figure 2, 4 and 6.**


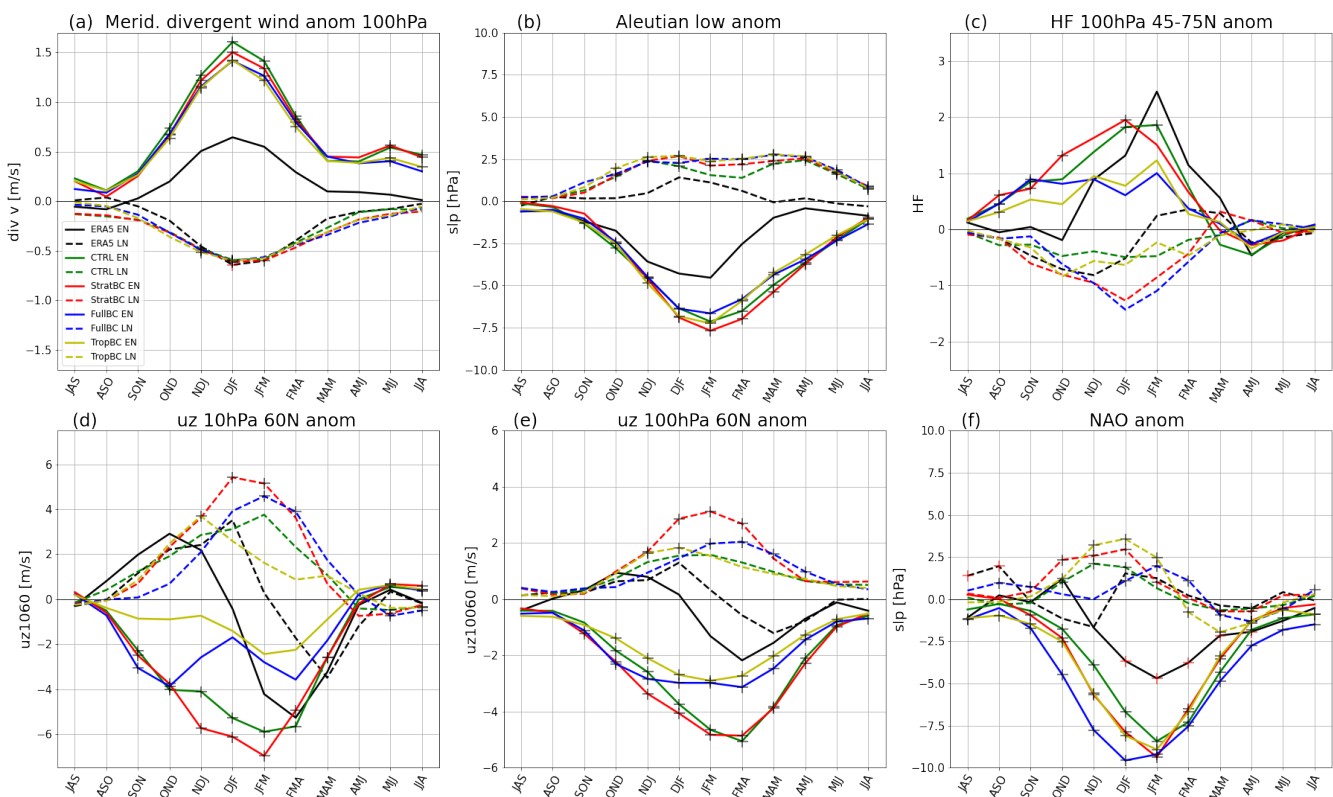

**Figure 2. Progression of anomalies from the ENSO region to the stratospheric polar vortex and NAO, for the model and ERA5. Timeseries uses three–month means, and black crosses indicate significance at p < 0.05 in the model, red crosses are used for ERA5. For model runs solid lines show 100 years of the El Niño run minus 100 years neutral run, dashed lines show 100 years of the La Niña run minus 100 years neutral run. ERA5 data is from 1950-2021 and shows the difference between a composite of 13 El Niño (black solid lines) or 16 La Niña years (black dashed lines), and 19 neutral years.**

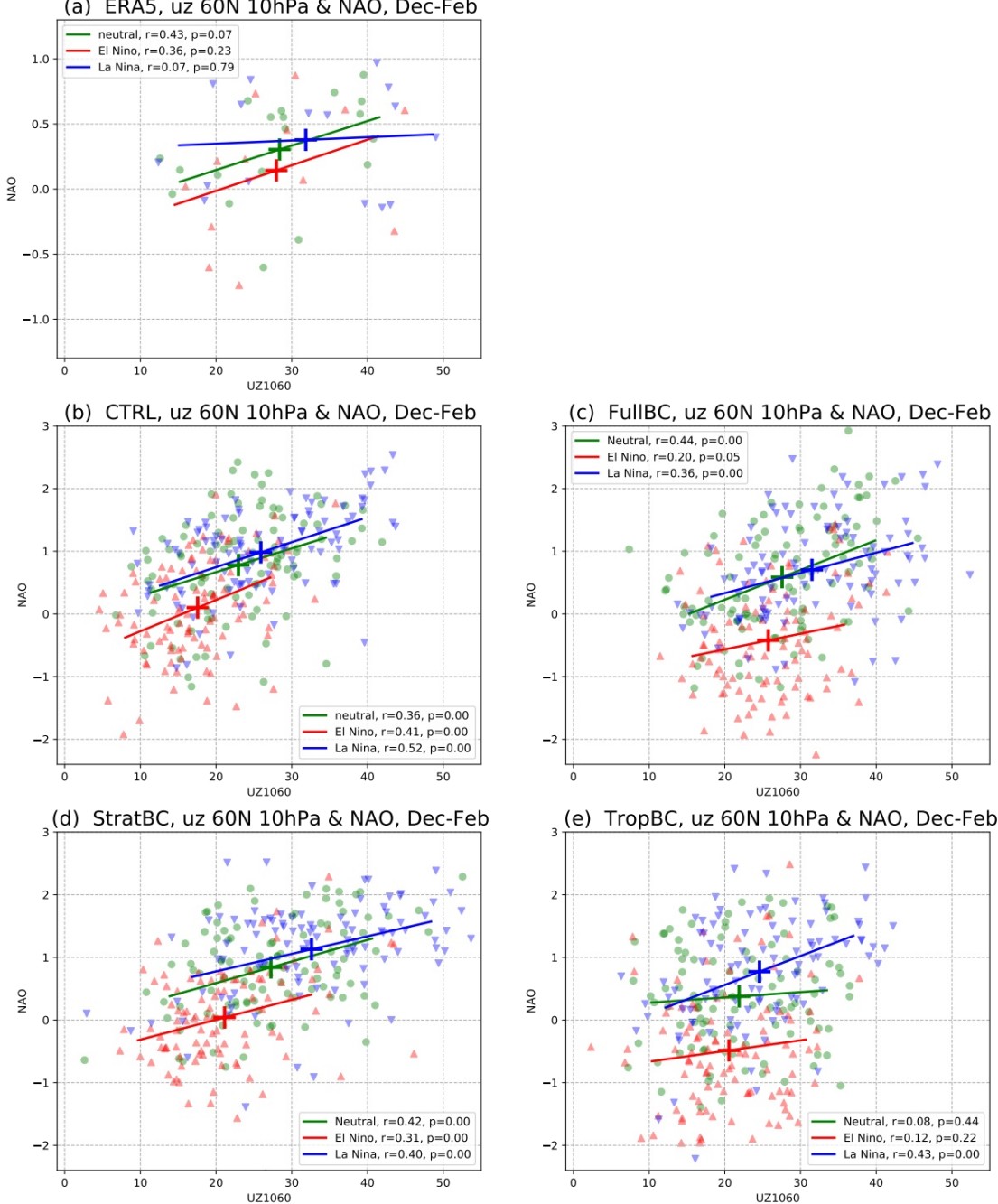

**Figure 3. Scatterplot with regression line of mean DJF values of uz at 60N 10hPa and the NAO index, for (a) ERA5 (1950-2021), (b) CTRL, (c) FullBC, (d) StratBC and (e) TropBC. Neutral years are green, El Niño years are red, and La Niña years are blue. The large crosses indicate the mean value for each El Niño/La Nina/neutral experiment. Correlation and p-values shown in the legend.**


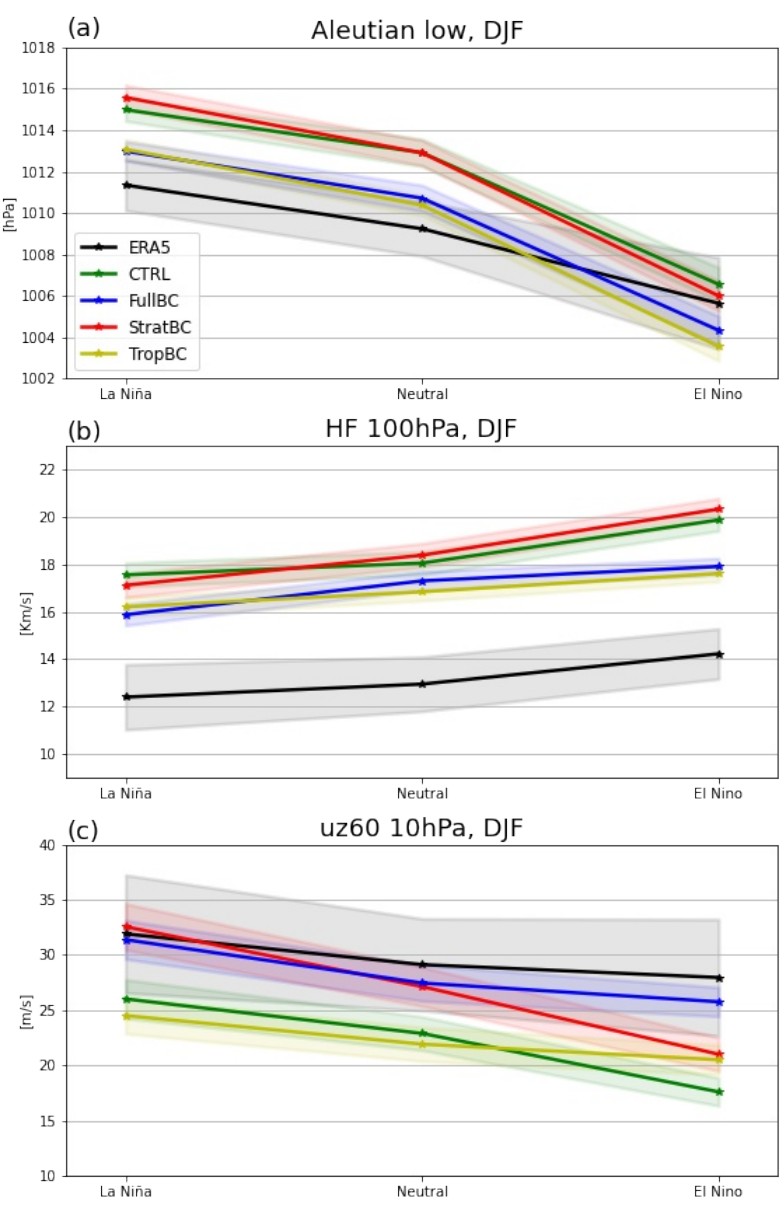

**Figure 4. DJF values in the model and ERA5 of (a) Aleutian Low Index, (b) heat flux between 45N–75N at 100 hPa and (c) UZ60 for La Niña, neutral and El Niño years. Shading shows 95% confidence interval.**


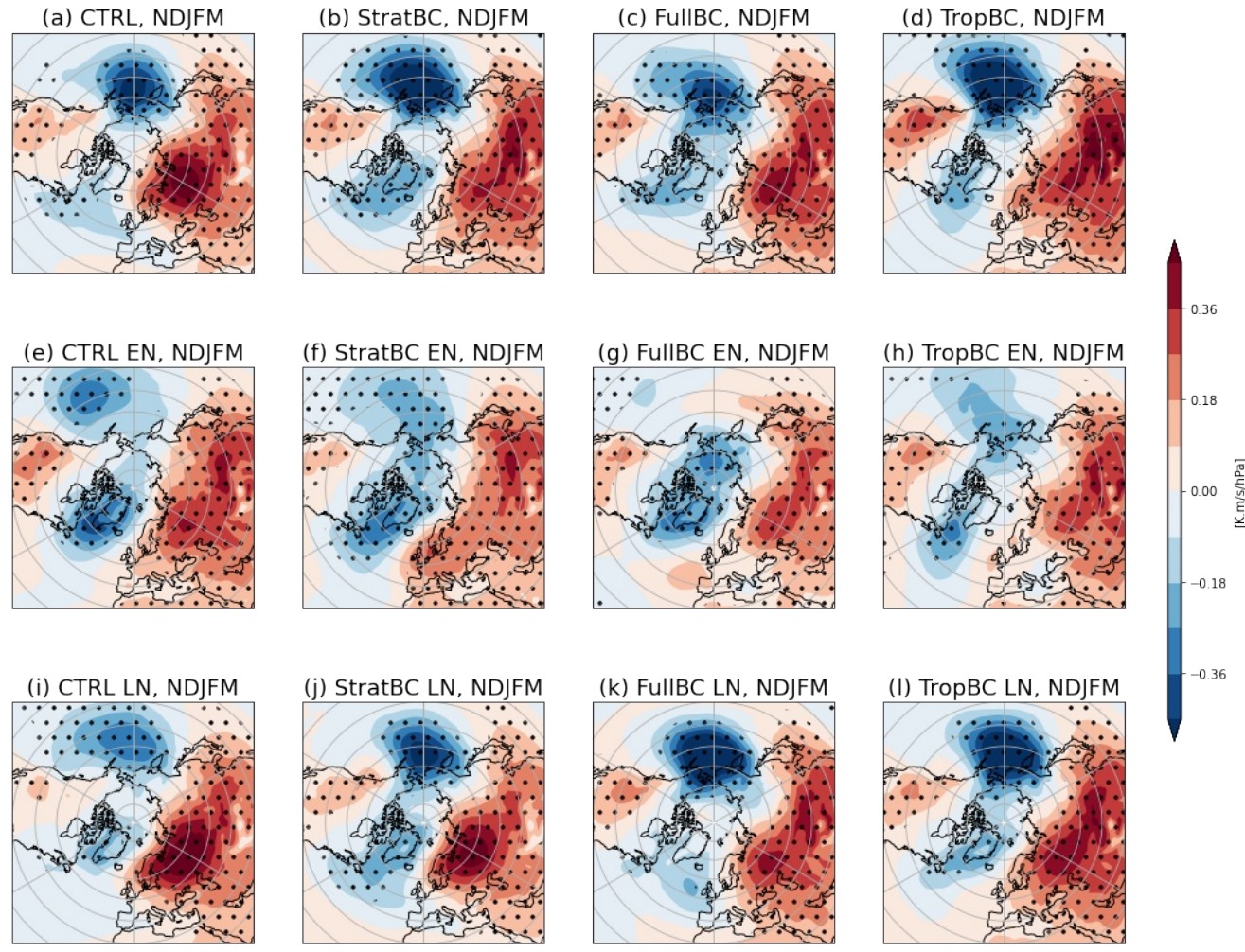

**Figure 5. Regression of monthly HF and monthly sea level pressure for extended winter months (Nov–Mar). Top row is neutral years, middle row is El Niño years, bottom row is La Niña years. Stippling indicates significance at p < 0.05.**

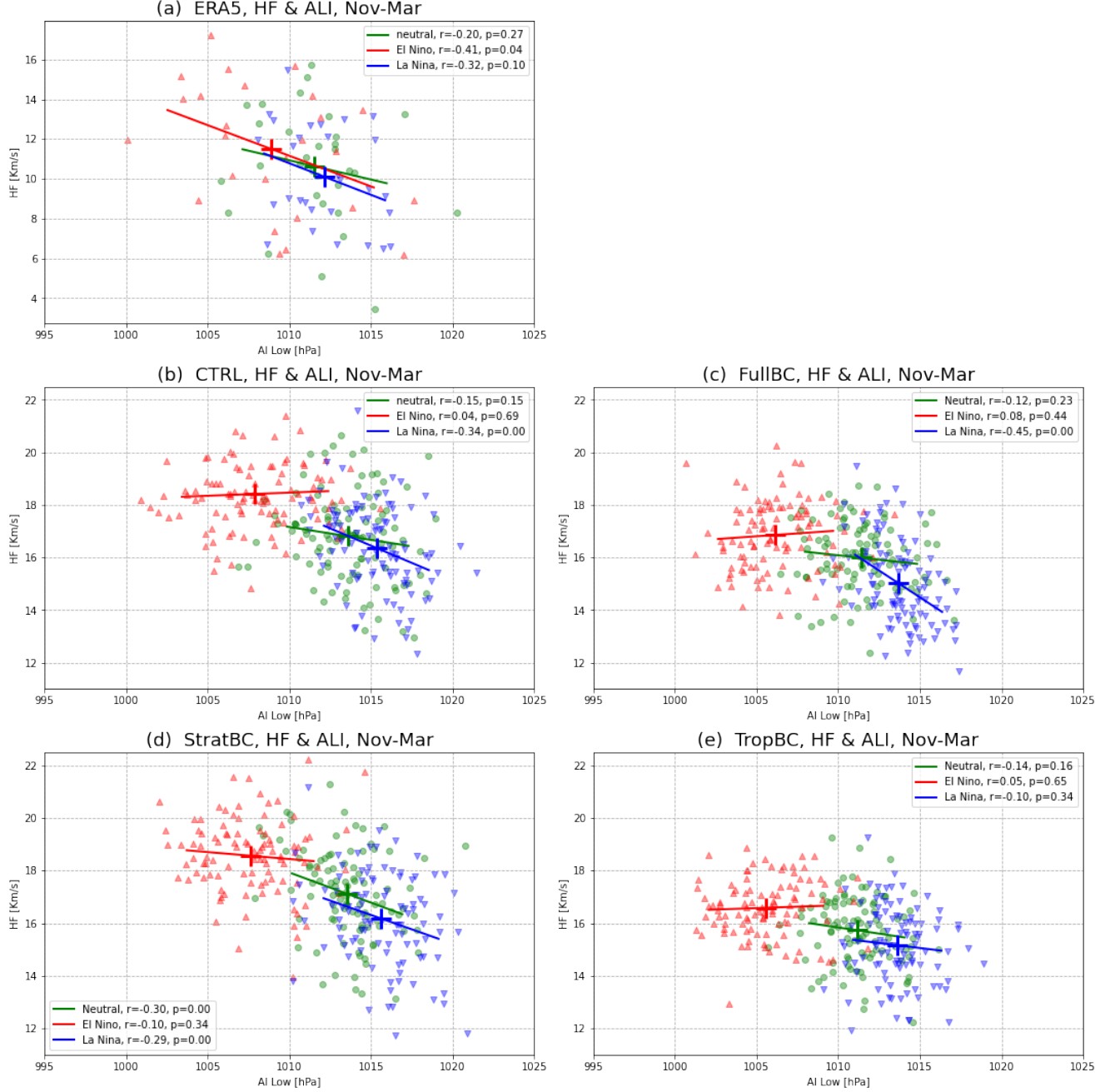

**Figure 6. Scatterplot with regression line of mean NDJFM values of the Aleutian Low Index and the heat flux between 45N–75N at**
**100 hPa, for (a) ERA5 (1950-2020), (b) CTRL, (c) FullBC, (d) StratBC and (e) TropBC. Neutral years are green, El Niño years are**
**red, and La Niña years are blue. Correlation and p-values shown in the legend.**