# Peer review of "Minimal impact of model biases on northern hemisphere ENSO teleconnections."

_Weather and Climate Dynamics, 2020_

## Referee Comment (RC1) · Anonymous Referee #1 · 23 Dec 2020

Minimal impact of model biases on northern hemisphere ENSO teleconnections
Nicholas L. Tyrell and Alexey Yu Karpechko

The authors have carried out a set of bias-correction experiments in the ECHAM 6 model investigating the role of basic-state model biases on the teleconnection between ENSO and the NAO. By imposing a climatologically varying set of fixed tendencies to the temperature and divergence fields in the troposphere, stratosphere, or both, they can partially correct model biases in different regions of the model. They then impose tropical pacific SST anomalies modeled after El Nino or La Nina states and analyse the impact on the NAO, considering established pathways for the teleconnection through

the stratospheric polar vortex.

They find that while the bias correction does affect the response of the polar vortex to ENSO, the NAO response itself is largely unaffected. They conclude from this that the teleconnection between ENSO and the NAO does not depend on the stratospheric pathway. Another interesting hypothesis presented is that the response of near-tropopause heat fluxes (indicative of vertically propagating waves) to the Aleutian low may saturate at large amplitudes.

This is an interesting methodology and potentially interesting set of results. However, I find the paper to be a bit light on detailed analysis of the runs, and I am not sure that the evidence presented supports the central conclusions:

1) While Figs 2c-e show that the zonal mean zonal winds at 60 N, 10 hPa respond much more in the Control and Strat cases than in the Trop and Full cases, there are several other possible interpretations of the fact that the NAO response to ENSO does not change in a consistent fashion in 2e beyond the conclusion that the stratosphere does not play a role in the ENSO-NAO teleconnection:

(a) the lower stratospheric response may not correspond to the 10 hPa polar vortex response.

(b) there may be compensating changes: for instance, the polar vortex response might have weakened in Trop and Full BC, but the sensitivity of the NAO to stratospheric variability may have increased.

(c) there may be significant non-linearities in the response of the NAO through the stratospheric teleconnection that complicate the interpretation of a time-mean sensitivity; not all winters will see a perturbation to the polar vortex.

2) Regardless, the bias correction acheived by this methodology is only partial: the biases that remain in the full bias-corrected run are in many regions larger-amplitude than those that have been corrected. That doesn't make these results irrelevant, but it

does narrow their applicability: it is not valid to conclude, for instance, that the response of the Aleutian low to ENSO is insensitive to *all* climatological biases (l196-197), only to the ones that were corrected here. Concluding statments in lines 211-212 and 224-226 should be similiarly qualified. The results found here are really only relevant to the biases that have in fact been corrected.

3) Closely related to 1 and 2; one should be cautious about inferring too much about the observed ENSO-NAO teleconnection on the basis of a single model result, especially if the bias-corrected NAO response to ENSO does not agree well with observations.

In my opinion the paper would have greater impact if some further analyses were performed (see below for details); I believe these would constitute major revisions. At a minimum, however, these three concerns need to be addressed in the text, and conclusions should be tempered to match the level of the evidence presented.

General comments:

Nature of biases that are corrected:

There could be considerably more detail given regarding the nature of the biases that are corrected. The RMSE plots in Fig 1 give a sense of their relative magnitude, but not of their structure. What do the zonal mean biases look like? What about the stationary wave features in the upper troposphere? There has been some success in understanding the ENSO to polar vortex teleconnection in terms of linear interference ideas; this might shed a lot more light on what's happening in these runs. These details would make the results much more useful.

Details of teleconnection:

Some of the concerns listed above could be addressed by further analyses. To me the most essential to address are

1) The reanalysis results are not shown in Fig. 2. Where the bias correction does modify the quantities that have been shown here, does it move the response more

closely to observations? Fig. 3 indicates there are meaningful differences between the modelled and observed teleconnection.

2) The lower stratospheric response should be shown.

Beyond these essential points, Fig. 4 is a bit distinct from the previous figures in that it presents an event-based perspective, instead of a time-mean response. The motivation for this isn't entirely made clear in the text, but the approach could be extended to further address the concerns raised above. e.g. (a) plotting the heat fluxes against SLP anomalies might confirm the hypothesis that there is a non-linear saturation of the upward part of the teleconnection, and (b) the response of the NAO to stratospheric sudden warmings in each model run could shed light on the strength of this coupling in each run.

Specific comments

Methods: Am I correct to infer that the runs have fixed climatological SSTs in all cases?

l 110: This seems overly optimistic to me - I see meaningful improvements in mid-latitudes, but pretty the improvements over the pole are minimal except in the upper stratosphere. There is a moderate improvement in the lower polar stratosphere (200-100 hPa) in the full bc run. Arguably the tropospheric bias correction degrades the mid-stratosphere.

l145-146: 'PV response not affected by biases' this needs to be explained more carefully - the response does depend on the bias corrections, although perhaps what the authors mean is that the differences in the response can be fully explained by the differences in the heat flux anomalies?

Fig 3. Does the shading show the standard deviation, or a confidence interval? It is notable that even the bias corrected model does not capture the non-linear nature of HF teleconnection. Also, a bar chart might be more appropriate here.

l197: The HF anomalies are likely connected to the SLP anomalies, but it's not clear

that they are driven by the SLP biases, especially when the active bias correction is imposed in T and divergence.

---

## Short Comment (SC1) · 5 Jan 2021

Hi,

I find your study interesting. I would like to mention a few papers you might find interesting too.

A couple recent studies highlight a varying ENSO teleconnection in the North Atlantic area from November through March:

Intraseasonal Effects of El Niño–Southern Oscillation on North Atlantic Climate

https://journals.ametsoc.org/view/journals/clim/31/21/jcli-d-18-

0097.1.xml?tab_body=fulltext-display

Importance of Late Fall ENSO Teleconnection in the Euro-Atlantic Sector

https://journals.ametsoc.org/view/journals/bams/99/7/bams-d-17-
0020.1.xml?tab_body=fulltext-display

Also, SST (or atmospheric heating) anomalies in other tropical ocean basins that co-vary with ENSO could play an important role in the evolution of the teleconnections from autumn to winter (also the first one above):

Separating the Indian and Pacific Ocean impacts on the Euro-Atlantic response to ENSO and its transition from early to late winter

https://journals.ametsoc.org/view/journals/clim/aop/JCLI-D-20-0075.1/JCLI-D-20-
0075.1.xml?rskey=Vu4nRs&result=6

The role of an Indian Ocean heating dipole in the ENSO teleconnection to the North Atlantic European region in early winter during 20th century in Reanalysis and CMIP5 simulations

https://journals.ametsoc.org/view/journals/clim/aop/jcliD200269/jcliD200269.xml?rskey=dBg|

Cheers,

Martin

---

## Referee Comment (RC2) · Anonymous Referee #2 · 10 Jan 2021

By using a two-step bias-correction technique, the influence of climatological bias in ECHAM6 spectral atmospheric model on its simulation of the ENSO-EU teleconnection is studied. The authors show that the AL intensity responds to the ENSO SST anomalies regardless of its long-term bias. While the polar vortex anomalies are weakened in TropBC and FullBC experiments, the NAO index is quite similar between the original biased and artificially-corrected simulations. Their research also found that the wave flux related to the stratosphere anomalies become less sensitive to the AL variability when the troposphere BC is improved, which leading to a weaker polar vortex signal in the upper stratosphere. In general, there is no significant difference in the ECHAM6 model's simulated ENSO-EU teleconnection (NAO intensity) before and after the bias–correction technique is applied. The topic should be of interest to model developers and forecasters. However, some points worth careful consideration before publication are:

General comments:

My major concern is about the conclusion made by the authors "a stratospheric response is not necessary for the ENSO–North Atlantic teleconnection". Some questions worth consideration are:

(1) Is there any downward propagation of stratospheric signals in the Ctrl experiment? If the answer is false, and in all these correction experiments there is also no such signal seen, then in all the simulations there is no influence from the stratosphere. However, the only conclusion we can reach here based on the modeling results is that the ECHAM6 can simulation the ENSO-EU teleconnection either with/without strong stratosphere anomalies. But it is not appropriate to conclude that the stratosphere is not important at all given the evidence found in OBS and many other studies.

(2) If all the experiments are accompanied with stratosphere-troposphere coupling, why the NAO signals are comparable between the experiments with strong and weak (FullBC and TropBC) stratosphere anomalies? Does that mean the BC in FullBC and TropBC are more favorable for downward coupling?

When the authors convey the opinion "The stratosphere BC bias is not the determinative factor for the upward wave flux", a reminder pops out that some improvements are also embedded in the mid-latitude stratosphere of the FullBC and TropBC simulations. Moreover, I am not sure if the StratBC could have any influence on the downward process. To sum up, I recommend the authors to give some downward propagation analyses concerning the polar vortex anomalies in each modeling experiment.

I also have concerns about how representative UZ60 at 10hPa for the whole polar vortex strength?

[Figure]

Specific comments:

Line 218-220: what's the evidence of this speculation? The Aleutian Low climatology bias reduction is just one feature in TropBC experiment. One can not rule out the possibility that the BC flow change in other areas / over upper levels plays a more important role.

Line 85-95: please explain why a regression method is used for extracting SST forcing fields not composite? Or how similar are the SST anomalies pattern in the sensitivity experiment and the one in observation? My point is that the regression patterns are exactly asymmetric between the EN and LN phases, but not in observation. Care should be taken when comparing the model results with OBS in Figure 3 then.

Figure1: why gives the 2D distribution of SLP but not at higher levels? The most obvious correction appears on the mid- to high- troposphere/stratosphere as shown by Figure1(e) and 1(f). Are the bias structure and corresponding improvement share a barotropic structure with the SLP field? If not, it's better to add more plots.

Line 110: It may be necessary to point out that in StratBC experiment, no significant improvement is seen below 20hPa (midlat) & 50hPa (polar), although the correction is applied from 100hPa.

Technical corrections:

Line 31-33: There is a paper by Mezzina et al (2020) titled "Multi-model assessment of the late-winter extra-tropical response to El Niño and La Niña" pointed out that the ENSO-forced SLP signal over the North Atlantic is kind of different from the NAO.

Line 195: "Hence, we find that climatological biases do not significantly affect the response of the Aleutian low to Rossby wave forcing, or the polar vortex to upward planetary wave forcing": upward planetary wave forcing does change because of TropBC correction, and the polar vortex intensity is highly corrected with that wave forcing. The authors might want to say "the StratBC climatological biases over the polar region is

not important", but the description used is overly general and might be misunderstood. Please improve that.

Figure 3(b) : misused units marked as [m/s]

Figure 2(a) legend: Strat->StratBC, Full->FullBC, Trop->TropBC

Figure 3(a) legend: Strat_BC->StratBC, Full_BC->FullBC, Trop_BC->TropBC

---

## Author Comment (AC1) · 16 Feb 2021

Dear referees,

Thank you for the thorough and thoughtful review or out paper. Our responses are in green. We will also include a "track changes" version of the revised paper.

Kind regards,

Nicholas and Alexey

**Anonymous Referee #1**

Minimal impact of model biases on northern hemisphere ENSO teleconnections

Nicholas L. Tyrrell and Alexey Yu Karpechko

The authors have carried out a set of bias-correction experiments in the ECHAM6 model investigating the role of basic-state model biases on the teleconnection between ENSO and the NAO. By imposing a climatologically varying set of fixed tendencies to the temperature and divergence fields in the troposphere, stratosphere, or both, they can partially correct model biases in different regions of the model. They then impose tropical pacific SST anomalies modeled after El Nino or La Nina states and analyse the impact on the NAO, considering established pathways for the teleconnection through the stratospheric polar vortex.

They find that while the bias correction does affect the response of the polar vortex to ENSO, the NAO response itself is largely unaffected. They conclude from this that the teleconnection between ENSO and the NAO does not depend on the stratospheric pathway. Another interesting hypothesis presented is that the response of near-troposphere heat fluxes (indicative of vertically propagating waves) to the Aleutian low may saturate at large amplitudes.

This is an interesting methodology and potentially interesting set of results. However, I find the paper to be a bit light on detailed analysis of the runs, and I am not sure that the evidence presented supports the central conclusions:

1) While Figs 2c-e show that the zonal mean zonal winds at 60 N, 10 hPa respond much more in the Control and Strat cases than in the Trop and Full cases, there are several other possible interpretations of the fact that the NAO response to ENSO does not change in a consistent fashion in 2e beyond the conclusion that the stratosphere does not play a role in the ENSO-NAO teleconnection:

We do not think the stratosphere doesn't play any role - its importance has been shown in a similar model (Manzini et al 2006) - but for our experiments the strength of the polar vortex response does not appear to control the strength of the NAO (or NAO-like) response. We have made this distinction more clear in the text. Our result also appears consistent with that by Bell et al. (2009) who also found that for strong ENSO events the tropospheric teleconnections dominate. However, we also agree that the additional analysis you suggested below is interesting and have included more results.

(a) the lower stratospheric response may not correspond to the 10 hPa polar vortex response.

We have included the 100hPa response in Figure 2, and also the full zonal mean El Nino and La Nina response in supplementary figures. We are confident that the 10hPa UZ is a reliable indicator of vortex strength for our experiments.

(b) there may be compensating changes: for instance, the polar vortex response might have weakened in Trop and Full BC, but the sensitivity of the NAO to stratospheric variability may have increased.

(c) there may be significant non-linearities in the response of the NAO through the stratospheric teleconnection that complicate the interpretation of a time-mean sensitivity; not all winters will see a perturbation to the polar vortex.

To explore the suggestions in (b) and (c) we have included a scatter plot of uz 60N 10hPa and the NAO index. For this figure we chose to show the variability within each experiment (i.e. 100 DJF means for CTRL EN, Neutral and LN) to understand the time-mean sensitivity. The figure shows that there is the expected relationship between UZ60 and the NAO within each experiment - a weaker vortex is associated with a more negative NAO. There is an indication that this relationship is also apparent between the El Nino, and Neutral and La Nina years (with a much smaller signal between the Neutral and La Nina experiments). However, the large variability within each experiment means that the forced difference is relatively small. Obviously, these plots do not elucidate causality in the relationship, but we believe they show that the polar vortex response is not the primary cause of the ENSO-EU teleconnection. The effect is relatively small compared to variability, and hence, different polar vortex responses between the bias correction experiments does not translate neatly into different magnitudes of the NAO response. Also the correlation between U10 and NAO are of similar magnitude among experiments, implying there is no increase in sensitivity as suggested in point (b).

2) Regardless, the bias correction achieved by this methodology is only partial: the biases that remain in the full bias-corrected run are in many regions larger-amplitude than those that have been corrected. That doesn't make these results irrelevant, but it does narrow their applicability: it is not valid to conclude, for instance, that the response of the Aleutian low to ENSO is insensitive to *all* climatological biases (l196-197), only to the ones that were corrected here. Concluding statements in lines 211-212 and 224-226 should be similarly qualified. The results found here are really only relevant to the biases that have in fact been corrected.

Thank you, this is a good point, and we have amended the text to include additional qualifiers.

3) Closely related to 1 and 2; one should be cautious about inferring too much about the observed ENSO-NAO teleconnection on the basis of a single model result, especially if the bias-corrected NAO response to ENSO does not agree well with observations.

Thank you for the observation. We have attempted to be clear in our conclusions about the limitations of the study.

In my opinion the paper would have greater impact if some further analyses were performed (see below for details); I believe these would constitute major revisions. At a minimum, however, these three concerns need to be addressed in the text, and conclusions should be tempered to match the level of the evidence presented.

**General comments:**

**Nature of biases that are corrected:**

There could be considerably more detail given regarding the nature of the biases that are corrected. The RMSE plots in Fig 1 give a sense of their relative magnitude, but not of their structure. What do

the zonal mean biases look like? What about the stationary wave features in the upper troposphere? There has been some success in understanding the ENSO to polar vortex teleconnection in terms of linear interference ideas; this might shed a lot more light on what's happening in these runs. These details would make the results much more useful.

We have included additional plots showing the biases and corrections in supplementary figures, showing the zonal winds, zonal mean temperature, and geopotential height at 300hPa.

**Details of teleconnection:**

Some of the concerns listed above could be addressed by further analyses. To me the most essential to address are

1) The reanalysis results are not shown in Fig. 2. Where the bias correction does modify the quantities that have been shown here, does it move the response more closely to observations? Fig. 3 indicates there are meaningful differences between the modelled and observed teleconnection.

We have added the ERA5 data to Fig. 2, reproduced below. There are, indeed, some very large differences between the observations and the model results, however there are also differences in the forcing of the model, i.e. a constant forcing of only the SSTs in the tropical Pacific, compared to the a composite of events which varies seasonally, also includes the SST response of other ocean basins, and is comprised of a fairly low number of events. While this complicates comparison between model and observations we do appreciate the need for a reference state, so we have included the ERA5 results and also discussed the complications involved

2) The lower stratospheric response should be shown.

We have included the 100hPa UZ in Figure 2, and also show lat-height plots of the bias corrections, and the ENSO response as supplementary figures

Beyond these essential points, Fig. 4 is a bit distinct from the previous figures in that it presents an

event-based perspective, instead of a time-mean response. The motivation for this isn't entirely made clear in the text, but the approach could be extended to further address the concerns raised above. e.g. (a) plotting the heat fluxes against SLP anomalies might confirm the hypothesis that there is a non-linear saturation of the upward part of the teleconnection, and (b) the response of the NAO to stratospheric sudden warmings in each model run could shed light on the strength of this coupling in each run.

Regarding a), we have looked at plots of SLP and heat flux. They show that there's an indication of a non-linear saturation, as suggested by the regression maps, but it's not a conclusive result. We will include the plots in supplementary figures for completeness. We have also motivated Figure 4 (now Fig 5) further.

Regarding b), we are currently working on a separate manuscript to look at the SSW responses within the experiments; we believe this question is distinct from a seasonal mean perspective pursued in the current manuscript and therefore it deserves a separate publication.

**Specific comments**

Methods: Am I correct to infer that the runs have fixed climatological SSTs in all cases?

That is correct, we have specified that in the text.

l 110: This seems overly optimistic to me - I see meaningful improvements in mid-latitudes, but the improvements over the pole are minimal except in the upper stratosphere. There is a moderate improvement in the lower polar stratosphere (200-100 hPa) in the full bc run. Arguably the tropospheric bias correction degrades the mid-stratosphere.

The line *"In FullBC the UZ RMSE is reduced throughout the atmosphere with reductions of over 50% in the upper troposphere–lower stratosphere region",* was meant to just be describing the mid-latitudes, and this has been clarified in the text. However, it could be that tropospheric bias-corrections are responsible for a degraded polar mid-stratosphere in the TropBC, and this has been noted in the text.

l145-146: 'PV response not affected by biases' this needs to be explained more carefully - the response does depend on the bias corrections, although perhaps what the authors mean is that the differences in the response can be fully explained by the differences in the heat flux anomalies?

We have clarified that statement: *"... the response of the polar vortex does not appear to be affected by biases in the strength of the vortex, and is instead fully explained by the heat flux response,"*

Fig 3. Does the shading show the standard deviation, or a confidence interval? It is notable that even the bias corrected model does not capture the non-linear nature of HF teleconnection. Also, a bar chart might be more appropriate here.

The shading is one standard deviation of CTRL and ERA5 (showing all experiments made the plot too hard to read). We would prefer this style of plot to a bar chart, since it's easier to see the change of each experiment relative to each other.

l197: The HF anomalies are likely connected to the SLP anomalies, but it's not clear that they are driven by the SLP biases, especially when the active bias correction is imposed in T and divergence.

We think the differences in the regression maps in Figure 4 (now Fig 5), along with the improvements in SLP biases when the troposphere is corrected (Fig1) show a strong connection, however, it is true that causality is difficult to confirm without more direct experiments, so we have discussed this in the conclusions.

**Anonymous Referee #2**

By using a two-step bias-correction technique, the influence of climatological bias in ECHAM6 spectral atmospheric model on its simulation of the ENSO-EU teleconnection is studied. The authors show that the AL intensity responds to the ENSO SST anomalies regardless of its long-term bias. While the polar vortex anomalies are weakened in TropBC and FullBC experiments, the NAO index is quite similar between the original biased and artificially-corrected simulations. Their research also found that the wave flux related to the stratosphere anomalies become less sensitive to the AL variability when the troposphere BC is improved, which leading to a weaker polar vortex signal in the upper stratosphere. In general, there is no significant difference in the ECHAM6 model's simulated ENSO-EU teleconnection (NAO intensity) before and after the bias–correction technique is applied. The topic should be of interest to model developers and forecasters. However, some points worth careful consideration before publication are:

**General comments:**

My major concern is about the conclusion made by the authors "a stratospheric response is not necessary for the ENSO–North Atlantic teleconnection". Some questions worth consideration are:
(1) Is there any downward propagation of stratospheric signals in the Ctrl experiment? If the answer is false, and in all these correction experiments there is also no such signal seen, then in all the simulations there is no influence from the stratosphere. However, the only conclusion we can reach here based on the modeling results is that the ECHAM6 can simulation the ENSO-EU teleconnection either with/without strong stratosphere anomalies. But it is not appropriate to conclude that the stratosphere is not important at all given the evidence found in OBS and many other studies.

We have clarified some statements about the importance of the stratosphere, we should be more clear in saying that we think it *is* an important part of the ENSO-EU teleconnections (e.g. as shown by Manzini et al 2006 with a very similar model to the one we used), but for our experiments the strength of the polar vortex response does not appear to control the strength of the NAO (or NAO-like) response. Our result also appears consistent with that by Bell et al. (2009) who found that for strong ENSO events the tropospheric teleconnections dominate. We have also tested the downward propagation within each experiment in Figure 3. Scatterplots of UZ60 vs NAO shows there is no evidence that the stratosphere-troposphere coupling has changed in the bias correction BC experiments.

(2) If all the experiments are accompanied with stratosphere-troposphere coupling, why the NAO signals are comparable between the experiments with strong and weak (FullBC and TropBC) stratosphere anomalies? Does that mean the BC in FullBC andTropBC are more favorable for downward coupling?

The downward coupling does not change significantly between the experiments, so it appears the magnitude of the NAO response is not directly tied to the magnitude of the stratospheric response.

When the authors convey the opinion "The stratosphere BC bias is not the determinative factor for the upward wave flux", a reminder pops out that some improvements are also embedded in the mid-latitude stratosphere of the FullBC and TropBC simulations. Moreover, I am not sure if the StratBC could have any influence on the downward process. To sum up, I recommend the authors to give some downward propagation analyses concerning the polar vortex anomalies in each modeling experiment.

Thank you for the suggestion. As mentioned we have included plots showing the stratosphere-troposphere coupling (Supp figure 4). The downward propagation appears weaker than in observations, but we do not see major differences between the experiments. We have also included

further analysis of the relationship between the vortex and the NAO (Figure 3).

I also have concerns about how representative UZ60 at 10hPa for the whole polar vortex strength?

We have included UZ60 at 100hPa in figure 2, and have shown latitude-height plots to more fully show the vortex response to ENSO (Supp Figure 5), and also the biases and their corrections (Supp figure 1). We are confident that UZ60 at 10hPa is representative of the vortex strength.

**Specific comments:**
Line 218-220: what's the evidence of this speculation? The Aleutian Low climatology bias reduction is just one feature in TropBC experiment. One can not rule out the possibility that the BC flow change in other areas / over upper levels plays a more important role.

We do agree that there could be other explanations, which is why we cautiously present this as a speculation. Our reasoning is based on a few things; the role of the Aleutian Low in the ENSO-Stratosphere connection, an analysis of the different bias corrections showing that a difference in this area (with FullBC and TropBC, different to StratBC and CTRL), the regressions (fig 5) suggesting the importance of the Aleutian Low in wave forcing and also changes in that relationship between the experiments. We're unsure if it's possible to further strengthen the speculation within the current experimental framework, but we think it's potentially an interesting feature, so we chose to discuss it

Line 85-95: please explain why a regression method is used for extracting SST forcing fields not composite? Or how similar are the SST anomalies pattern in the sensitivity experiment and the one in observation? My point is that the regression patterns are exactly asymmetric between the EN and LN phases, but not in observation. Care should be taken when comparing the model results with OBS in Figure 3 then.

We acknowledge the ENSO forcing experiments are quite idealized, and therefore care should be taken when comparing directly to observations. Since our main aim was to test the influence of model biases, decisions were made to simplify the experiments, this included using a regression (and hence a symmetric EN/LN signal), and also using a constant forcing, rather than a more realistic seasonally varying forcing. We have added to the text to emphasize this.

Figure1: why gives the 2D distribution of SLP but not at higher levels? The most obvious correction appears on the mid- to high- troposphere/stratosphere as shown by Figure1(e) and 1(f). Are the bias structure and corresponding improvement share a barotropic structure with the SLP field? If not, it's better to add more plots.
SLP biases indicate distribution of atmospheric mass and therefore they are critical for analysis of the bias correction improvements. We have added supplementary plots to show more detail of the biases and corrections. We show the zonal mean zonal wind and temperature, as well as the geopotential height at 300hPa. The latter shows that the structure in the SLP field is barotropic.

Line 110: It may be necessary to point out that in StratBC experiment, no significant improvement is seen below 20hPa (midlat) & 50hPa (polar), although the correction is applied from 100hPa.
Yes, agree, text changed

Technical corrections:Line 31-33: There is a paper by Mezzina et al (2020) titled "Multi-model assessment of the late-winter extra-tropical response to El Niño and La Niña" pointed out that the ENSO-forced SLP signal over the North Atlantic is kind of different from the NAO.
Thank you, that's a very interesting paper, we have added the reference.

Line 195: "Hence, we find that climatological biases do not significantly affect the response of the Aleutian low to Rossby wave forcing, or the polar vortex to upward planetary wave forcing": upward planetary wave forcing does change because of TropBC correction, and the polar vortex intensity is highly corrected with that wave forcing. The authors might want to say "the StratBC climatological

biases over the polar region is not important", but the description used is overly general and might be misunderstood. Please improve that.
Agree, text changed

Figure 3(b) : misused units marked as [m/s]
Fixed

Figure 2(a) legend: Strat->StratBC, Full->FullBC, Trop->TropBC
Fixed

Figure 3(a) legend: Strat_BC->StratBC, Full_BC->FullBC, Trop_BC->TropBC
Fixed

---

## Author Comment (AC2) · 16 Feb 2021

Hello Martin,

Thank you for your interest in our work, the papers you mentioned are very interesting. For this study we found them relevant to consider when comparing our idealized model results to observations. We are currently looking more closely at how the bias-corrections effect variability, so they will be perhaps more useful for that work too.

Kind regards, Nicholas and Alexey
* * *
[Figure]

2020.

---

## Author Response (AR2)

**Reviewer #1**

Minor points:

The last sentence in the abstract "implying that for strong ENSO events a stratospheric pathway may not be necessary for the ENSO-North Atlantic teleconnection" : recommend change "necessary" to other words such as "dominant" since all the experiments still have a stratospheric pathway although maybe not as important as the tropospheric one.
We agree, changed

Line 130: reorder the Supplementary Figures to let Figure 1 appear first in the text.
The supplementary figures have been reordered

Line 144: "La Niño" -> "La Niña" - fixed

Line 145: "ad" -> "and" - fixed

Line 187: "La Nina" -> "La Niña" - fixed

Figure 3: How about in OBS? Or in other words, does the author believe that the weak role played by the stratosphere is a character of the current models or also true in OBS? Better mention this somewhere in the text.
We have added ERA5 to Figure 3 (also to Figure 5), and discuss it in the text. We think our model runs show evidence that for strong ENSO events the tropospheric teleconnection is dominant, which we point out in the discussion, i.e. *"This result appears consistent with that by Bell et al. (2009) who also found that for strong ENSO events the tropospheric teleconnections dominate.",* although we don't think the observational evidence we present in this paper necessarily confirms that result.

Line 193: suggest cite the corresponding figure before "because of …." - agree, added "Figure 4 c"

Supplementary Figure 5: suggest unify the subplots' Y coordinates - agree, merged into a single colorbar

Figure 5 and Supplementary Figure 5: It looks that the HF in FullBC and TropBC ( Figure 5(k) and 5(l) ) is more sensitive to the AL change than those in CTRL and StratBC ( Figure 5(i) and 5(j) ), which is different from the results shown in Supplementary Figure 5. I notice that the season used for calculation is also different, which may cause the inconsistency? - Agree, we have used NDJFM for both plots for consistency (Supp Fig 5 has moved to Fig 6)

**Reviewer #2**

The authors have carried out a set of bias-correction experiments in the ECHAM 6 model investigating the role of basic-state model biases on the teleconnection between ENSO and the NAO. By imposing a climatologically varying set of fixed tendencies to the temperature and divergence fields in the troposphere, stratosphere, or both, they can partially correct model biases in different regions of the model. They then impose tropical pacific SST anomalies modeled after El Nino or La Nina states and analyse the impact on the NAO, considering established pathways for the teleconnection through the stratospheric polar vortex.

They find that while the bias correction does affect the response of the polar vortex to ENSO, the NAO response to ENSO itself is largely unaffected. They conclude from this that, in the model, the teleconnection between ENSO and the NAO does not depend on the stratospheric pathway, and that it is not sensitive to the model biases corrected for by the bias correction methodology employed. Another interesting hypothesis presented is that the response of near-tropopause heat fluxes (indicative of vertically propagating waves) to the Aleutian low may saturate at large amplitudes.

I have a few questions and concerns outlined below; my main criticism is that the authors should discuss their results more quantitatively in the context of the observed teleconnection. The text could also have used another read through to catch typos.

If these concerns are meaningfully addressed the paper should meet the standard for publication.

**Major comments**

Structure of Biases

I have the same criticism of this revised paper as I did of the discussion paper: the authors discuss the effects of model bias on the ENSO/NAO teleconnection without clearly showing or discussing the detailed structure of the biases. The supplementary figures (1 through 3) are far more meaningful than the profiles of RMSE shown in Figure 1. They should be included in the main text in addition (or in lieu of) the current panels a-f of Figure 1 and the discussion in 3.1 should be rewritten in terms of the structure of the temperature and wind biases that are corrected.

In particular, there are strong cold biases (I assume what is shown is model - reanalyis, though this is not specified) at high latitudes near and above the tropopause evident in supplemental figure 2. This is included in the TropBC run, but the bias may be better thought of as a lower-stratospheric bias. This could be indicative of an extratropical tropopause that is too high in the model. The TropBC run reduces the high latitude temprature anomalies at the expense of introducing cold anomalies in the tropical upper troposphere. The wind biases that are corrected are in thermal wind balance with these anomalies (as expected), but this makes sense of the significant reduction in RMSE found in the mid-latitude stratosphere in TropBC and FullBC.

Thank you for the constructive criticism regarding this section, and apologies for not correcting in the first round of reviews. We have replaced the RMSE profiles in Figure 1 with zonal mean plots from the supplementary figures, as they are much more descriptive. We have re-written section 3.1, outlining the specific biases, particularly those relevant for ENSO teleconnections, and details of where the biases have been reduced in the different experiments.

Structure of ENSO response

The differences between the modeled and observed teleconnections are very substantial, particularly with regards to the stratospheric pathway of the teleconnection. This is material to the central conclusions of the manuscript with regards to the relative importance of the tropospheric pathway in this model: the model is not properly capturing the stratospheric pathway in its basic configuration or in the bias correction runs. This is particularly evident with regards to the response of the heat flux metric (panel 2c), which is highly nonlinear in the observations, with a positive response to either signed tropical Pacific SST anomalies. This leads to a highly nonlinear stratospheric response (panels d and e), and an NAO response with a seasonal structure that more closely resembles the stratospheric seasonal cycle than the seasonal cycle of the tropical divergence response.

Statements made in lines 163 and 207-8 are far too speculative and the hypothesis that this non-linearity is consistent with sampling variability should be quantitatively tested. This can be done by subsampling periods from the model runs as long as the observational record to determine if something like the observational signal could have occurred plausibly by chance within the context of the models variability. Given the large structural differences seen in Fig. 2 this comparison needs to be done more quantitatively.

These differences also need to be explicitly discussed both in section 3 and in the conclusions.

We have approached this in two ways. Firstly, since the manuscript was first submitted more ERA5 data has become available, with a back extension to 1950. We have included this extra data to increase the number of ENSO events we can compare to. We find the HF response to be less non-linear (perhaps a detailed study of the ERA5 Back Extension ENSO response would be of interest, but beyod the scope of this paper). Secondly, as suggested, we have subsampled our model runs. We show the subsampling results using the JFM value of HF, since this is the period with the greatest difference between model and reanalysis. We find the observed values are within the upper adjacent values, indicating the observational values are within the sampling uncertainty of all model runs. Therefore, while the reviewer is right pointing to the differences between observed and simulated teleconnections, our tests indicate that there is no evidence that the models do not capture the observed stratospheric pathway of ENSO teleconnections. Observational records are simply too short to constrain the magnitude of ENSO teleconnection even with respect to the sign of the response.

**Specific comments:**

l 34: This isn't consistent with Figure 2; the NAO response is negative, not positive. Similar non-linearities are seen in HF and stratospheric responses.
That was worded badly, corrected

l 158: 'almost opposite' this is ambiguous and confusing - the text could be read to mean that the La Nina response is also an increase in HF but with CTRL and StratBC runs showing half as strong a response as Full BC and TropBC.
Agree, the sentence was re-written.

l 162-3: what is the 'canonical' response here? the observations?
This was poorly worded and has been changed

l 273: I find supplemental figure 5 quite helpful in supporting this argument - I think it's worth showing in the main text.
Agree, it has been moved to the main text

Methodology:

Are the temperature and divergence bias correction tendencies computed from a nudging run in which only temperature and divergence are relaxed, or ar they taken from a run in which all four fields are nudged?
All four fields - divergence, vorticity, temperature, log surface pressure - were used in the nudging stage, then just divergence and temperature for the bias correction. We have added some clarification to Section 2. This setup was found experimentally, to have the greatest reduction in biases without introducing additional errors. Because the key goal of the procedure is to reduce biases we believe our approach is well justified.

Fig. 4: Since we are comparing magnitudes of responses in different runs, the shading should show a confidence interval (which is relevant for comparison), not a measure of variability (which is not). Changed

Typos

l57: affect - fixed

l81: nudged - fixed

l145 symmetric and actually - fixed

l163: response. - fixed

l185-187: Please clarify - text has been rewritten

l190: which effect? - The effect of a weaker or stronger vortex on the NAO. Text added

l222: its - fixed

l269: closer to observations. - fixed